# Widely Targeted Metabolomics and Network Pharmacology Reveal the Nutritional Potential of Yellowhorn (*Xanthoceras sorbifolium* Bunge) Leaves and Flowers

**DOI:** 10.3390/foods13081274

**Published:** 2024-04-21

**Authors:** Haojie Sha, Shouke Li, Jiaxing Li, Junying Zhao, Dingding Su

**Affiliations:** 1Peking University Institute of Advanced Agricultural Sciences, Weifang 261325, China; haojie.sha@pku-iaas.edu.cn (H.S.); 13253621162@163.com (J.L.); junying.zhao@pku-iaas.edu.cn (J.Z.); 2Shandong Woqi Agricultural Development Co., Ltd., Weifang 262100, China; lishouke@163.com

**Keywords:** yellowhorn, leaf and flower, widely targeted metabolomics, volatile and non-volatile metabolites, network pharmacology

## Abstract

Yellowhorn (*Xanthoceras sorbifolium* Bunge) is a unique oilseed tree in China with high edible and medicinal value. However, the application potential of yellowhorn has not been adequately explored. In this study, widely targeted metabolomics (HPLC-MS/MS and GC-MS) and network pharmacology were applied to investigate the nutritional potential of yellowhorn leaves and flowers. The widely targeted metabolomics results suggested that the yellowhorn leaf contains 948 non-volatile metabolites and 638 volatile metabolites, while the yellowhorn flower contains 976 and 636, respectively. A non-volatile metabolite analysis revealed that yellowhorn leaves and flowers contain a variety of functional components beneficial to the human body, such as terpenoids, flavonoids, alkaloids, lignans and coumarins, phenolic acids, amino acids, and nucleotides. An analysis of volatile metabolites indicated that the combined action of various volatile compounds, such as 2-furanmethanol, β-icon, and 2-methyl-3-furanthiol, provides the special flavor of yellowhorn leaves and flowers. A network pharmacology analysis showed that various components in the flowers and leaves of yellowhorn have a wide range of biological activities. This study deepens our understanding of the non-volatile and volatile metabolites in yellowhorn and provides a theoretical basis and data support for the whole resource application of yellowhorn.

## 1. Introduction

Yellowhorn (*Xanthoceras sorbifolium* Bunge), an oil seed tree belonging to the *Sapindaceae* family, is widely distributed in northern China [1]. Yellowhorn is known as “northern camellia oleifera” in China. The oil content in a yellowhorn kernel is more than 50%, and its oil is nutritious and edible [2,3]. Yellowhorn oil is classified as a traditionally consumed edible oil by the National Health Commission of the People’s Republic of China (https://slps.jdzx.net.cn/xwfb/gzcx/PassFileQuery.jsp accessed on 28 September 2020), and the Standard of Yellowhorn Oil was published by the National Food and Strategic Reserves Administration of the People’s Republic of China (LS/T 3265-2019, http://www.lswz.gov.cn/ accessed on 13 June 2020).

The fruit yield of yellowhorn is meager and is dubbed “the tree of a thousand flowers but one fruit” [4]. Despite the higher yield of its leaves and flowers, there is a significant gap in resource utilization compared with prevalent commercial-scented tea. For example, the flowers of *Citrus aurantium* L., *chrysanthemi* flos, and *Lonicerae japonicae* flos are popular among consumers due to their abundant resources, delicious taste, and high medicinal value [5]. A study conducted by Xiao et al. confirmed that the flavonoids in yellowhorn flower tea had potential pharmaceutical value against oxidative stress [5]. The flower and calyx of yellowhorn contain baicalin, which has antipyretic, sleep-inducing, antispasmodic, and anti-tumor effects [6].

Additionally, the leaf of yellowhorn contains some trace elements, such as zinc, manganese, and iron, which the human body needs. The leaf of yellowhorn is an excellent raw material for tea because its protein content is higher than that of black tea and its caffeine content is similar to that of flower tea [6]. The leaf and flower of yellowhorn could serve as resources to produce functional food. However, data support for the development of corolla leaves and flowers into new resource foods still needs to be completed.

Previous research primarily concentrated on isolating and identifying bioactive substances from leaves and flowers or their extracts. For instance, Xiao et al. found that five new barrigenol-type triterpenoids from leaves have anti-tumor activities [7]. The ethanolic extract of yellowhorn leaves is rich in active phenolic substances, such as quercetin, kaempferol-3-O-rutinoside, myricetin-3-O-rutinoside, epicatechin, and catechin, which could be primary components of the anti-neuroinflammatory effect and could potentially be a natural therapeutic remedy for neurodegenerative diseases [8]. According to Xiao et al.’s research, the main constituents of the yellowhorn flower are myricitrin, isoquercitrin, quercitrin, 4-O-β-D-glucopyranosyl-trans-*p*-coumaric acid, caffeic acid (1-hydroxyl-4-O-β-D-glucopyranosylprenyl)-ester, pinonesinol, and syringic acid [5]. Currently, a comprehensive understanding of the composition of yellowhorn leaf and flower metabolites is scant.

Benefiting from the rapid development of metabolomics technology, widely targeted metabolomics could analyze and characterize the quality and composition of food by identifying and quantifying thousands of known metabolites [9]. To improve the resource utilization of yellowhorn leaves and flowers, a thorough understanding of their metabolites is required. In this study, the non-volatile metabolites in yellowhorn leaves and flowers were investigated using ultra-performance liquid chromatography–tandem mass spectrometry (UPLC-MS/MS). The volatile metabolites were analyzed using head space solid-phase microextraction (HS-SPME) and gas chromatography–mass spectrometry (GC-MS). The findings of this study can help identify the functional components that confer the health benefits of the leaves and flowers of yellowhorn, as well as provide valuable information for the further development of new resources and functionally fortified food.

## 2. Materials and Methods

### 2.1. Materials

Yellowhorn is grown in Weifang (longitude: 119.1, latitude: 36.62), China. The leaves and flowers were provided by Shandong Woqi Agricultural Development Co., Ltd. (Weifang 262100, China).

### 2.2. Yellowhorn Flower and Leaf Sample Pretreatment

#### 2.2.1. UPLC-MS/MS Sample Pretreatment

Based on the method described by Chen et al. [10], the leaves and flower samples were freeze-dried (Scientz-100F, SCIENTZ, Ningbo, China) and ground to powder (MM 400, Retsch, Haan, Germany, 30 Hz, 1.5 min). Then, the leaf (Leaf 1, Leaf 2, Leaf 3) and flower (Flower 1, Flower 2, Flower 3) powder (50 mg for each) were dissolved in 1.2 mL of a 70% methanol solution, respectively. The metabolites were extracted through vortexing (a total of 6 times, vortexed for 30 s every 30 min). After centrifugation (12,000 rpm, 3 min), the supernatant was filtrated (0.22 μm microporous membrane) and stored in an injection bottle for subsequent UPLC-MS/MS analysis. To ensure the reproducibility of the mass spectrometric (MS) results, an equal mixture of the flower and leaf samples was used as a quality control (QC) sample.

#### 2.2.2. GC-MS Sample Pretreatment

The volatile compounds of the leaf and flower were extracted via head space solid-phase microextraction (HS-SPME) [11]. The samples of leaves and flowers were ground to a powder in liquid nitrogen. Then, the powder (500 g) was transferred to a 20 mL head space vial (Agilent, Palo Alto, CA, USA) with a NaCl-saturated solution. For HS-SPME, each vial was first heated to 60 °C for 5 min and then exposed to a 120 m DVB/CWR/PDMS fiber (Agilent) for 15 min.

### 2.3. UPLC-MS/MS Analysis

The non-volatile metabolites in yellowhorn leaves and flowers were identified using the UPLC-MS/MS system (UPLC, SHIMADZU Nexera X2, Kyoto, Japan, MS, Applied Biosystems 4500 Q TRAP, Thermo, Waltham, MA, USA). The UPLC conditions were as follows: the column was an Agilent SB-C18 (Agilent, Santa Clara, CA, USA, 2.1 mm × 100 mm, 1.8 µm) and the mobile phase included solvent A (pure water with 0.1% formic acid) and solvent B (acetonitrile with 0.1% formic acid). A gradient program was set up at a 0–9 min linear gradient from 5% to 95% B; the sample was held at 95% B for 1 min, returned to 5% for 10–11 min, and held at 5% B for 14 min (total 25 min). The sample injection settings were as follows: the flow velocity was 0.35 mL/min; the column temperature was 40 °C; and the sample volume was 4 μL.

The MS conditions were as follows: the electrospray ionization (ESI) temperature was 550 °C; the ion spray voltage was 5500 V (positive ion mode) and −4500 V (negative ion mode), respectively; the ion source gas I, gas II, and curtain gas were set at 50, 60, and 25 psi, respectively; and the collision-activated dissociation was high.

### 2.4. GC-MS Analysis

The volatile metabolites of yellowhorn leaves and flowers were identified using the GC-MS system (GC, 8890; MS, 7000D, Agilent, Santa Clara, CA, USA). The UPLC conditions were as follows: the column was an Agilent DB-5MS (30 m × 0.25 mm × 0.25 μm); the carrier gas was helium (99.999%); the velocity was 1.2 mL/min; and the injector temperature was kept at 250 °C. The temperature program was set as follows: hold at 40 °C for 3.5 min; 3.5–9.5 min linear gradient from 40 to 100 °C; 7 °C/min to 180 °C. At 25 °C/min to 280 °C, hold at 180 °C for 5 min.

The MS conditions were as follows: the electron energy was 70 eV; the ion source temperature was 230 °C; the quadrupole mass detector temperature was 150 °C; and the transfer line temperature was 280 °C. The scan mode was set to the selected ion monitoring mode.

### 2.5. Data Processing and Analysis

The HPLC-MS data (Appendix A) were analyzed using Analyst 1.6.3, and the GC-MS data (Appendix A) were analyzed using MassHunter (Agilent, Santa Clara, CA, USA). The results are displayed in Appendix A. In the metabolomics analysis, unsupervised principal component analysis (PCA) was performed on the leaf and flower samples to reveal the non-volatile or volatile metabolite differences between leaves and flowers. The differential metabolites were screened using supervised multiple regression orthogonal partial least-squares discriminant analysis (OPLS-DA). The screening criteria were variable importance in projection (VIP) value ≥ 1 and Log2|fold change| ≥ 1. Differential metabolites between groups were mapped to the Kyoto Encyclopedia of Genes and Genomes (KEGG) database to obtain detailed pathway information.

### 2.6. Network Pharmacological Analysis

The potential targets of the non-volatile metabolites in the leaves and flowers of yellowhorn were predicted using the Super-PRED database (https://prediction.charite.de/index.php accessed on 21 December 2023) based on their chemical structures. Hyperlipidemia, neurodegenerative diseases, prostatitis, and enuresis-associated targets were verified using the Disease Target Network database (DisGeNET, https://www.disgenet.org/ accessed on 21 December 2023). The interaction network between disease-associated targets and non-volatile metabolites was constructed and visualized using Cytoscape 3.9.1 software (Cytoscape Consortium, San Diego, CA, USA). The interaction between disease-associated targets and non-volatile metabolites was analyzed using the STRING database (https://string-db.org accessed on 21 December 2023). The protein–protein interaction (PPI) network was visualized using Cytoscape 3.9.1 software.

## 3. Results and Discussion

### 3.1. Overview of Non-Volatile Metabolites in Leaves and Flowers

Metabolomics was performed to obtain a comprehensive metabolite composition of the leaf and flower of yellowhorn in this study. A total of 948 and 976 non-volatile metabolites were identified in the leaf and flower, respectively (Figure 1A,B and Appendix A), and the primary and secondary classifications of these metabolite types are shown in Appendix A. The top five most abundant metabolite classes in the leaf were flavonoids (21.41%), lipids (15.82%), phenolic acids (15.40%), amino acids and derivatives (8.76%), and alkaloids (8.65%), while the top five most abundant metabolite classes in the flower were flavonoids (22.64%), phenolic acids (16.08%), lipids (15.16%), amino acids and derivatives (8.61%), and alkaloids (8.50%). These results indicated insignificant differences in the types of non-volatile compounds between the leaf and flower samples, echoing Xiao et al.’s study [5]. However, flowers contained higher levels of flavonoids, lipids, nucleotides, and derivatives in terms of relative content (Figure 1C, Appendix A). The abundant metabolite types and contents in the leaf and flower of yellowhorn suggest that they might have potentially beneficial effects on human nutrition and health.

The PCA between the leaf and flower samples and the quality control samples is shown in Figure 1D. The principal components 1 and 2 were 67.71% and 14.16%, respectively. The leaf samples were separated from the flower samples, indicating a significant difference between them. The correlation between samples was assessed using the Pearson correlation coefficient R. The correlation between the two samples is stronger when the R2 is close to 1. The data showed that the difference between the leaf and flower samples was significantly greater than those among replicates of the same tissue (Appendix A). This indicated that the metabolite data obtained were relatively accurate and reproducible and could be used for subsequent analyses.

### 3.2. Identification of Differential Non-Volatile Metabolites

The differential metabolites between the leaf and flower of yellowhorn were identified through OPLS-DA (Appendix A). As shown in Figure 1D, we identified 492 differential metabolites between the leaves and flowers. Compared with the leaves, the numbers of down- and up-regulated metabolites were 165 and 327 in the flowers, respectively. The differential metabolites were mainly flavonoids, lipids, phenolic acids, alkaloids, nucleotides and derivatives, and amino acids and derivatives (Appendix A).

The KEGG database is the primary public pathway database for understanding metabolism, membrane transport, signal transduction, and cell cycle [12]. To identify the main pathways of metabolite accumulation in the leaf and flower of yellowhorn, the differential metabolites were annotated and enriched for the comparison group and divided into different pathways in this study. The KEGG annotation results revealed that the differential metabolites between the leaf and flower were involved in 79 pathways, mainly in plant metabolism and the biosynthesis of secondary metabolites (Appendix A). The KEGG enrichment analysis is illustrated using bubble plots (Figure 1E). The metabolic pathways related to flavonoid biosynthesis, nucleotide metabolism, and purine metabolism were significantly enriched between the leaf and flower of yellowhorn. The results indicated that there were different metabolite profiles in the leaf and flower of yellowhorn.

### 3.3. Analysis of Non-Volatile Metabolites in the Leaves and Flowers of Yellowhorn

The medicinal value of yellowhorn has been documented in the Chinese Pharmacopoeia. The employment of the wood, leaf, seed, kernel, flower, and carpophore of yellowhorn as medicative materials incorporates a history extending thousands of years in China [13]. Yellowhorn has multiple active ingredients, including triterpenoids, flavonoids, alkaloids, phenolic acids, and other substances [6,13]. Therefore, the active substances with potential medicinal and nutritional value were further analyzed. The high-content metabolites in the leaf and flower are listed in Appendix A.

#### 3.3.1. Terpenoids

The terpenoid metabolites in the leaf and flower of yellowhorn are shown in Figure 2A. In total, 11 terpenoids were screened in the plant sample. Terpenoids comprise several subclasses, including monoterpenes, sesquiterpenes, diterpenes, triterpenes, and tetraterpenes [14]. In particular, triterpenes are the main bioactive substances of yellowhorn due to their diversity, content, and pharmacological activity [6]. Triterpenes usually exist in free form and in the form of glycosides or esters combined with sugars in yellowhorn [13]. This study found five triterpenes (compounds **3**–**7**), and their contents in the leaf were higher than those in the flower. To date, no studies have shown that these compounds have been found in yellowhorn. Compound **3** (pomolic acid) is a ursane-type pentacyclic triterpenoid with multiple effects, including neuroprotective, free radical scavenging, antioxidant, anti-inflammatory, and antiproliferative activity [15]. Compound **4** (corosolic acid), also known as phytoinsulin, has the pharmacological effects of hypoglycemic, anti-tumor, and anti-inflammatory activities and the prevention of oxidative stress [16]. The natural compound is currently marketed as a nutritional dietary supplement in the USA [17]. Compound **5** (jujubogenin) is a dammarane-type tetracyclic triterpene saponin. Compound **6** (hederagenin) is an oleane-type pentacyclic triterpene saponin [18]. Compound **7** (30-norhederagenin) is a pentacyclic triterpenoid found in *Paeonia* [19]. Wu et al. found anti-inflammatory effects in *Paeonia* extracts containing 30-norhederagenin [20]. The triterpene saponin (compound **8**, medicagenic acid-3-O-glucuronide-28-O-xylosyl(1,4)-[apiosyl(1,3)]-rhamnosyl(1,2)-arabinoside) was significantly higher in the leaves than that in the flowers. The saponin element of this triterpene saponin is medicagenic acid, an oleane-type pentacyclic triterpenoid. Compound **1** (loliolide), a monoterpenoid substance with high content in both the leaf and flower of yellowhorn, has anti-inflammatory and neurological disease-preventing effects [21,22].

#### 3.3.2. Flavonoids

Currently, numerous flavonoids have been isolated and identified from yellowhorn [13]. Flavonoids are found in a large number of dietary plants and herbs, either in the free form (aglycones) or bound to sugars [23,24]. Structurally diverse flavonoids can be classified into subclasses: flavones, flavanols, flavanones, and flavonols [24]. Recent studies have shown that flavonoids have multiple benefits in human health, including anti-cancer, antioxidant, and anti-inflammatory activity, cardiovascular protection, and gut health [23,24,25]. Flavones and flavonols were the primary flavonoid metabolites in the yellowhorn flower (Figure 2B), whose basic backbones included chrysoeriol, diosmetin, hispidulin, kaempferol, luteolin, and quercitrin. Compounds **29**–**39** and **43**–**46** (chrysoeriol-7-O-glucoside, diosmetin, diosmetin-7-O-galactoside, diosmetin-7-O-glucoside, hispidulin, hispidulin-7-O-glucoside, 6-C-MethylKaempferol-3-glucoside, luteolin-3′-O-glucoside, luteolin-4′-O-glucoside, luteolin-7-O-gentiobioside, 3,5,4′-Trihydroxy-7-methoxyflavone, kaempferol-3-O-galactoside-4′-O-glucoside, kaempferol-3-O-glucoside, kaempferol-3-O-sambubioside, and amoenin) have higher content in the yellowhorn flower compared to the leaf. Numerous studies have indicated that natural plant flavonoids (diosmetin, hispidulin, and kaempferol) have a variety of nutritional properties [26,27,28,29]. Diosmetin exhibits a wide range of potential applications in the treatment of various disorders, as it possesses anti-inflammatory, antioxidant, antimicrobial, antilipolytic, and analgesic activities [26]. Research evidence shows that hispidulin has anti-inflammatory, antifungal, antiplatelet, anticonvulsant, antiosteoporotic, and, notably, anti-cancer activities [28]. Kaempferol has been found to have considerable neuroprotective effects, regulating various inflammation-related signaling pathways and promoting dopamine release in the brain [28]. This indicates that the yellowhorn flower may be considered a pleiotropic functional food.

Compared with the yellowhorn flower, the proportion of flavonols (compound **20**–**23**, catechin, epicatechin, epigallocatechin, and gallocatechin) was high in the leaf. Catechins and their gallate esters have attracted much attention regarding their beneficial effect on human health. Green tea, a popularly consumed beverage, is rich in catechin. Green tea has been indicated to reduce the risk of many chronic diseases in epidemiological studies and clinical trials [30]. In addition, the derivatives of apigenin and quercetin were found in the leaf. A recent study found that Alzheimer’s disease may be averted by apigenin because it inhibits cellular aging processes [31]. Quercetin has a protective effect on the central nervous system, which reduces oxidative stress and neuroinflammation [32]. Hence, the development of yellowhorn leaf tea is also rewarding.

#### 3.3.3. Alkaloids

Alkaloids are heterocyclic structures containing one or more nitrogen atoms and are widely found in approximately 300 plant families [33]. Alkaloids have diverse biological and pharmacological activities, such as codeine for analgesia, aconitine for antiarrhythmic properties, and quinine for antimalarial activity [34]. As shown in Figure 2C, four kinds of alkaloids (compounds **70**, **79**, **89**, **90**) in the leaf and flower of yellowhorn have higher content. Compound **70** (trigonelline), a pyridine alkaloid, has the pharmacological properties of antiapoptotic, anti-inflammatory, antioxidant, anti-diabetic, and neuroprotective effects, which have the potential for cognition improvement [35]. Spermine (compound **83**) is naturally present in all living organisms and plays a vital role in synthesizing nucleic acids and proteins, gene expression, protein functions, and protection from oxidative stress [36]. Spermine is a polyamine, and polyamine supplementation may extend the life span in model organisms [37]. While polyamines have been reported to pose health hazards, such as lowering blood pressure and causing respiratory symptoms and nephrotoxicity, the work of del Rio et al. does not support this assertion [38]. Compound **90** (indole-3-carboxaldehyde) is a plumerane alkaloid, which was detected in high amounts in the leaf and flower samples in this study [39]. Moreover, another plumerane alkaloid (compound **89**, 1-methoxy-indole-3-acetamide) was found, which has not been previously reported in yellowhorn. In addition, serotonin (compound **86**, 5-hydroxytryptamine) is an important neurotransmitter in the brain. Our study found that its content in leaves is significantly higher than that in the flower. Serotonin regulates various behavioral and biological functions in the body, which play a role in the psychological processes of the central nervous system and peripheral tissues (bone and gut) [40].

#### 3.3.4. Lignans and Coumarins

Lignans, a kind of phenylpropanoid compound, are structurally complex bioactive polyphenolic phytochemicals formed by the coupling of two coniferyl alcohol residues and can be grouped into plant lignans and mammalian lignans based on their origin [41]. Lignans are widely distributed in nature and exhibit various biological characteristics, such as anti-tumor, antioxidant, antibacterial, and antiviral activities [42]. In the leaf and flower of yellowhorn, savinin, pinoresinol and its derivative, and dehydrodiconiferyl alcohol-4-O-glucoside were found (Figure 2D). Among them, pinoresinol can improve memory impairment through its effects on acetylcholinesterase and calcium influx [43].

Coumarins are a class of phenylpropanoid compounds with anti-cancer, anti-inflammatory, antimicrobial, and antithrombosis activities [44]. The coumarins in yellowhorn mainly included scopoletin, fraxin, isoscopoletin, isofraxetin, esculetin, and fraxetin, distributed mainly in the leaves, flowers, and husk [13]. In addition to these substances, scopolin and methylillicinone F were detected in the leaf and flower of yellowhorn in this study. Methylillicinone F has the highest content in the leaf and flower of yellowhorn (Figure 2D). However, its biological function has yet to be identified.

#### 3.3.5. Phenolic Acids

Phenolic acids can be divided into two major groups, hydroxycinnamic and hydroxybenzoic acid, which are derived from cinnamic and benzoic acid, respectively. As shown in Figure 2E, compared to the flower, two hydroxycinnamic acid derivatives (1-O-*p*-coumaroyl-β-D-glucose and *p*-coumaric acid-4-O-glucoside) had significantly higher content in the leaf. In addition, other hydroxycinnamic acid derivatives were found in the leaf and flower, such as 3-O-*p*-coumaroylquinic acid, 1-O-caffeoyl-β-D-glucose, 3-hydroxycinnamic acid, 2-hydroxycinnamic acid, and α-hydroxycinnamic acid. Our study found six new hydroxybenzoic acid derivatives, including 4-hydroxybenzoic acid, 3,4-dihydroxybenzoic acid (protocatechuic acid), 2,3-dihydroxybenzoic acid, 4-(3,4,5-trihydroxybenzoxy) benzoic acid, 2,5-dihydroxybenzoic acid, and 2,3,4-trihydroxybenzoic acid. Epidemiological and experimental evidence described the protective effects of phenolic acids in degenerative diseases such as cardiovascular, cancer, diabetes, and inflammation [45]. Phenolic acids are well recognized for their powerful antioxidant and anti-inflammatory actions [46].

#### 3.3.6. Amino Acids

Amino acids, the basic units of protein, are organic compounds containing amine (-NH_2_) and carboxyl (-COOH) functional groups. As shown in Figure 2F, the leaves and flowers of yellowhorn contain five essential amino acids, including lysine (compound **157**), tryptophan (compound **150**), leucine (compound **149**), phenylalanine (compound **156**), and valine (compound **151**). The leaves contained more leucine, valine, and lysine, while the flowers contained more tryptophan. Amino acids are not only constituents of proteins, but also precursors of metabolites. Arginine (compound **148**) and lysine, the common precursors of polyamine synthesis, are found in the leaves and flowers of yellowhorn. L-glutamic acid can be converted to L-glutamine under the catalysis of glutamine synthase. In addition, it can positively affect gut health by supporting the gut microbiome and gut mucosal wall integrity as well as modulating inflammatory responses [47].

#### 3.3.7. Nucleotides

Multiple physiological and biological functions of dietary nucleotides have been identified, including the modification of intestinal flora, a reduction in diarrheal episodes, the maintenance of normal growth and development in infants, the modulation of immune function, and gastrointestinal health [48]. As shown in Figure 2G, several nucleotides and derivatives were identified from the leaves and flowers of yellowhorn, including 2-aminopurine (compound **161**), guanosine (compound **162**), vidarabine (compound **163**), and adenosine (compound **164**). 2-Aminopurine has the function of treating lung cancer and pulmonary fibrosis [49]. Guanosine is a purine nucleoside with important functions. Evidence from rodent and cell models shows the neurotrophic and neuroprotective effects of guanosine, which alleviate symptoms associated with epilepsy, spinal cord injury, pain, mood disorders, and aging [50]. Vidarabine is a broad-spectrum antiviral agent and has been used to treat herpesvirus infections [51]. The computer results reported by Eissa et al. indicated that vidarabine was a potential SARS-CoV-2 nsp10 inhibitor [52]. Adenosine, another nucleotide, is a neurotransmitter of excitatory and inhibitory activity in the brain and is used to treat diseases of the central nervous system and the cardiovascular system [53].

In summary, the leaves and flowers of yellowhorn are rich in bioactive substances, such as terpenoids, flavonoids, alkaloids, phenolic acids, lignans, coumarins, phenolic acids, amino acids, and nucleotides. These components provide the potential to reduce inflammation, maintain intestinal health, and protect the central nervous system, proving that yellowhorn has great potential as a new food resource. This further increases the development and utilization value of yellowhorn.

### 3.4. Overview of Volatile Metabolites in the Leaf and Flower

Volatile substances are important compounds that contribute significantly to the aromatic properties of yellowhorn leaves and flowers, which were analyzed using GC-MS. As shown in Figure 3A, 638 and 636 volatile compounds were obtained from the leaf and flower samples, respectively. The relative abundance of volatiles in the leaf sample was in the following order: heterocyclic compounds (17.87%) > terpenoids (17.71%) > esters (15.51%) > hydrocarbons (11.13%) > ketones (9.72%) > aldehydes (8.15%) > alcohols (6.11%) > aromatics (5.96%) > acids (1.57%) > nitrogen compounds (1.41%) > phenols (1.25%) > sulfur compounds (1.25%) > amine (1.10%) > halogenated hydrocarbons (0.78) > others (0.47%). However, the relative abundance of volatiles in the flower sample was different from that in the leaf: heterocyclic compounds (17.45%) > terpenoids (17.29%) > esters (15.88%) > hydrocarbons (11.32%) > ketones (9.75%) > aldehydes (8.18%) > alcohols (6.29%) > aromatics (5.97%) > acids (1.57%) > nitrogen compound (1.42%) > phenols (1.25%) > sulfur compounds (1.25%) > amine (1.10%) > halogenated hydrocarbons (0.78) > others (0.47%). The data indicated no significant difference in the types of volatile between the leaf and flower samples.

To further investigate the differences in the volatile compounds in the leaf and flower samples, relative content analysis was performed. As shown in Figure 3B, it was found that the contents of the main volatile compounds (heterocyclic compounds, alcohols, terpenoids, and esters) in the leaf samples were higher than those in the flower samples. The leaves and flowers differed in terms of relative abundance of the volatiles (Figure 3C). The data showed that 2-furanmethanol had the highest relative content in both the leaf and flower, accounting for 12.43% and 14.15%, respectively. 2-furanmethanol has good antioxidant activity in beer, and its antioxidant activity and concentration showed a dose–dependent relationship [54]. Thus, the volatile compounds in the leaves and flowers might have a powerful antioxidant ability. 5-Ethyl-3-hydroxy-4-methyl-2(5H)-Furanone, also known as abhexone, has been identified as one of the volatile compounds responsible for the sweet caramel odor in coffee and strawberry vinegar [55]. Naphthalene, an unpleasant volatile, was identified as an aroma-active compound in green teas and white teas [56,57]. However, the contribution of volatile substances to flavor is usually not directly related to their abundance. Generally, those substances with a low odor threshold and high relative content tend to contribute more to the food flavor. 2-methyl-3-furanthiol is an important meat aroma compound in wine, coffee, sesame, and cooked meat because of its extremely low odor threshold (0.007 ppb) [58]. The formation of 2-methyl-3-furanthiol is related to the Maillard reaction [59]. Another volatile compound with an extremely low odor threshold is β-ionone (0.007 ppb), which is a major contributor to the flavor of green and black tea [60]. It can be produced either by enzymatic reactions during fermentation or thermal degradation during the green tea manufacturing process [61]. The abundance of these volatile components was higher in the leaves, which indicated that yellowhorn leaves can be developed as tea products.

### 3.5. Identification of Differential Volatile Metabolites

The PCA between the two samples is shown in Figure 4A. Principal components 1 and 2 explained 72.91% and 10.27% of the total variance, respectively. The leaf samples and flower samples were separated, indicating a significant difference in volatile compounds between the leaves and flowers. To further explore this difference, differential metabolites analysis was carried out. A total of 136 differential volatile compounds (117 down-regulated and 19 up-regulated) were found between the leaf and flower samples (Figure 4B). These differential metabolites were mainly heterocyclic compounds, terpenoids, hydrocarbons, esters, ketones, aldehydes, aromatics, alcohols, phenols, acids, halogenated hydrocarbons, and amines (Figure 4C), indicating that the aroma-active compounds between the leaf and flower samples differed.

Figure 4D shows the significant differential volatile compounds between the leaves and flowers of yellowhorn. Several compounds are found only in the flowers, such as methyl ester, (Z)-9-hexadecenoic acid, heneicosane, 6,10,14-trimethyl-, (E, E)-5,9,13-pentadecatrien-2-one, octadecane, 2-cyclopentylethanol, and methyl ester 9-octadecenoic acid (Z)-. Straight-chain hydrocarbons (heneicosane and octadecane) may derive from the wax component of the flower [62]. The low hydrocarbon content gives the typical rosaceous and fresh character of high-quality rose oil [63]. Methyl ester, (Z)-9-hexadecenoic acid, and methyl ester 9-octadecenoic acid (Z)- are types of esters. Esters usually have high odor threshold values, and they do not significantly contribute to the overall aroma of some foods [64]. Farnesyl acetone (6,10,14-trimethyl-, (E, E)-5,9,13-pentadecatrien-2-one) was reported to be present in teas, such as Pu’er tea and black tea [65,66]. Two pyrazines (acetylpyrazine and (2-methylpropyl)-pyrazine) were found in the leaf. Acetylpyrazine was confirmed as one important flavor substance contributing to the unique roast aroma of Dahongpao tea [67].

The above results indicate that there are various and abundant volatile substances in the leaves and flowers of yellowhorn, thus affecting the aroma quality. The aroma of the leaves and flowers is a combination of various flavor-active compounds. However, this investigation did not take the detected volatile chemicals’ odor threshold into account. In the future, research should focus on determining the association between volatile compounds and the odorant perception of humans.

### 3.6. Network Pharmacology of the Leaf and Flower of Yellowhorn

Yellowhorn is a unique oilseed tree species grown in China. It is an excellent species for use in desertification control, ecological greening, soil and water conservation, and saline–alkali land development in northern China due to its strong tolerance to drought, cold, and barren saline–alkali environments [68]. Its wood texture is rugged and beautiful and can be used as high-grade household building material. Due to its colorful, numerous flowers, it is known as the “Chinese cherry blossom” and has great ornamental value. Most current research mainly focuses on yellowhorn fruit as a bio-oil source and its resistance mechanism in response to drought and heat stress. The study of the flowers and leaves of the fruit needs to be more systematic and comprehensive. The leaves and flowers of yellowhorn have a certain medicinal and health value and are used to prevent and treat diseases. The plant can be made into tea, which has the effects of lowering blood lipids, blood pressure, blood glucose, and uric acid [6]. The National Health Commission of the People’s Republic of China (www.nhc.gov.cn accessed on 24 July 2023) has approved the kernels and leaves of the fruit as a new food. Hence, we assessed the potential of the leaves and flowers of yellowhorn for therapeutic applications using network pharmacology.

Yellowhorn has a central role in traditional Chinese and Mongolian medicinal therapies. The plant has positive effects on learning and memory and has the function of anti-inflammatory, anti-tumor, and anti-oxidative properties [6,69]. Therefore, four diseases (hyperlipidemia, neurodegenerative diseases, prostatitis, and enuresis) were selected for examination in a study of the therapeutic potential of yellowhorn leaves and flowers. In total, 33 and 34 non-volatile metabolites were collected from yellowhorn leaves and flowers (Appendix A). The targets of these compounds were 424 (leaf) and 382 (flower), respectively (Appendix A). As shown in Figure 5A–D, there were 64, 168, 11, and 1 intersection targets between the non-volatile metabolites in yellowhorn and hyperlipidemia, neurodegenerative diseases, prostatitis, and enuresis, respectively. After both metabolite targets and disease targets were imported into Cytoscape software 3.9.1, a “metabolite-target-disease” network was constructed (Figure 5E). The metabolites in the leaves and flowers of yellowhorn have a high potential in treating neurodegenerative diseases (NDs). The intersection between the metabolites in the leaf and flower and ND was 47 and 36, respectively (Figure 5F). The xanthoceraside present in yellowhorn has been reported to improve learning and memory impairment in mice with an intracerebroventricular injection of Aβ_1–42_ [70]. In our study, pomolic acid, kaempferol, catechin, epicatechin, epigallocatechin, gallocatechin, quercetin, and 5-hydroxytryptamine were found in the leaves and flowers of yellowhorn. These compounds have been reported to have anti-inflammatory and neuroprotective properties [21,28]. The major metabolites in the leaf and flower were identified through the “metabolite–target–disease” network. The top five metabolites in the flowers were zarzissine, 2-aminopurine, hispidulin, luteolin-3′-O-glucoside, and naringenin, and the top five metabolites in the leaves were jujubogenin, serotonin, tryptamine, L-lysine, and rhoifolin (Appendix A). Zarzissine is an alkaloid that has hypoglycemic and anti-inflammatory effects [71]. The biological activity of hispidulin, luteolin-3′-O-glucoside, and naringenin has also been widely reported, and these components have anti-inflammatory, antioxidant, antifungal, anti-cancer, and neuroprotective properties, as well as immunoregulatory activity [28,72,73]. Jujubogenin has been found in the leaves of yellowhorn, which has notable neuroprotective effects in dementia-related diseases, such as Alzheimer’s disease, via its anti-inflammation, anti-apoptosis, and antioxidant properties [74]. Therefore, yellowhorn is a native Chinese plant with promising applications as a medicinal and edible plant with the potential for whole-resource applications.

## 4. Conclusions

Yellowhorn is a woody oil plant unique to China that has great potential for exploitation. In this study, UPLC-MS/MS combined with HS-SPME-GC-MS was applied for the first time to comprehensively investigate and compare both non-volatile and volatile metabolite profiles in the leaves and flowers of yellowhorn. The findings revealed that the yellowhorn leaves contain 948 non-volatile and 638 volatile metabolites, while the flowers contain 976 and 636, respectively. The leaves and flowers have anti-inflammatory, antioxidant, and central nervous system protective effects due to the presence of these metabolites. The results indicate that the leaves and flowers of yellowhorn have potential for use as new food resources. This study is helpful for the exploitation of yellowhorn as a whole resource; however, research on the biological activity of the metabolites is limited. In the future, more studies are needed to investigate the formation mechanism and utilization of the functional and key flavor components of the leaves and flowers.

## Figures and Tables

**Figure 1 foods-13-01274-f001:**
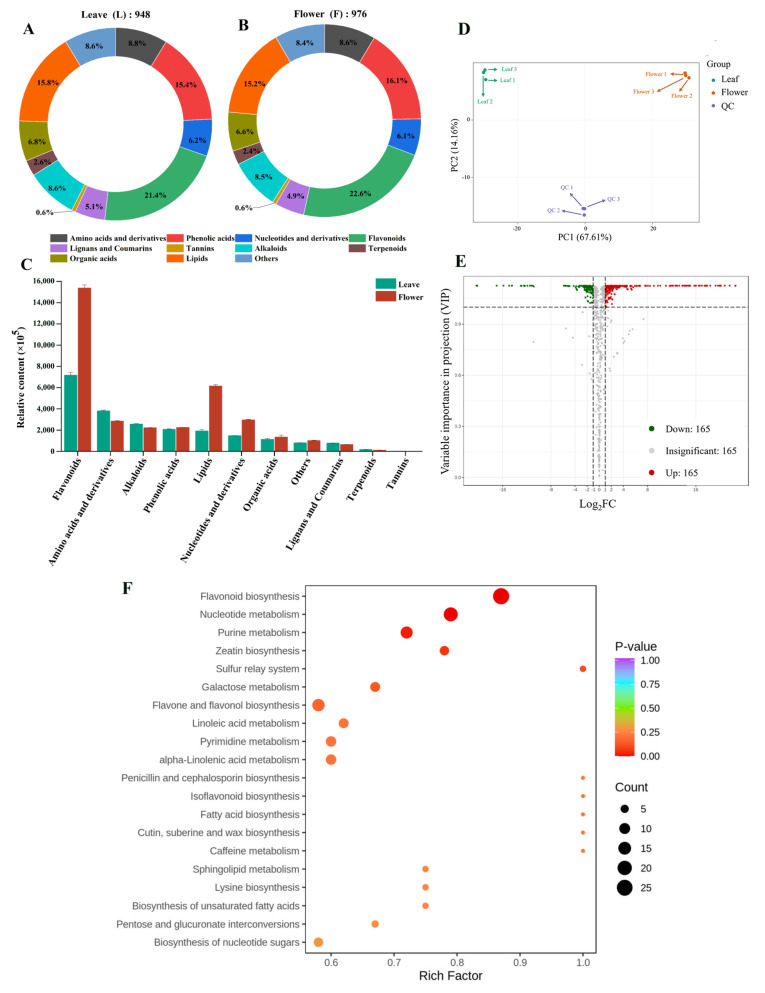
Non−volatile metabolites in yellowhorn leaves and flowers. (**A**,**B**) Proportion of different classes of non−volatile metabolites; (**C**) content comparison of different classes of non-volatile metabolites; (**D**) PCA score plots; (**E**) volcano plot of non-volatile metabolites; (**F**) KEGG annotations and enrichment results of the differentially expressed non-volatile metabolites.

**Figure 2 foods-13-01274-f002:**
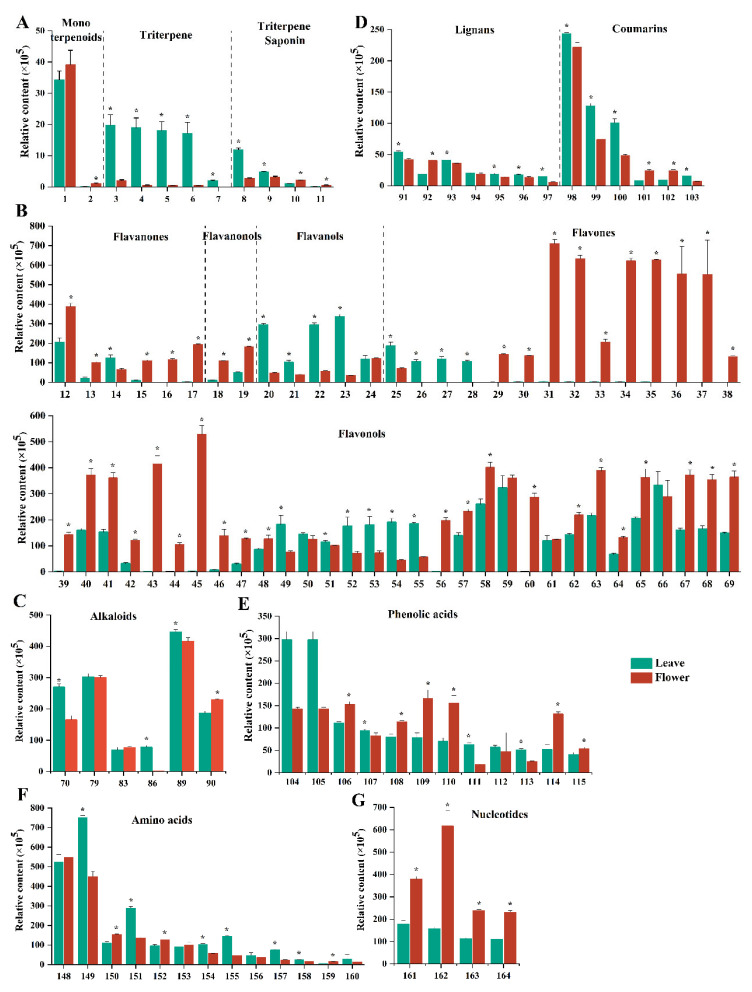
Relative content of non-volatile metabolites in yellowhorn leaves and flowers. (**A**) Mono terpenoids, triterpene, and triterpene saponin; (**B**) flavanones, flavanonols, flavanols, flavones, and flavonols; (**C**) alkaloids; (**D**) lignans and coumarins; (**E**) phenolic acids; (**F**) amino acids; (**G**) nucleotides. * indicates significant difference (*p* < 0.05).

**Figure 3 foods-13-01274-f003:**
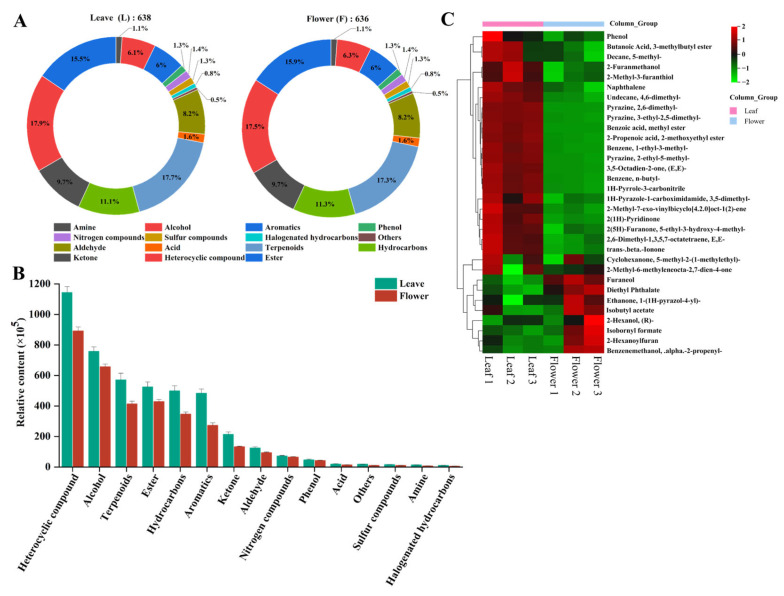
Volatile metabolites in yellowhorn leaves and flowers. (**A**) Proportion of different classes of volatile metabolites; (**B**) content comparison of different classes of volatile metabolites; (**C**) heat map of hierarchical cluster analysis for volatile metabolites.

**Figure 4 foods-13-01274-f004:**
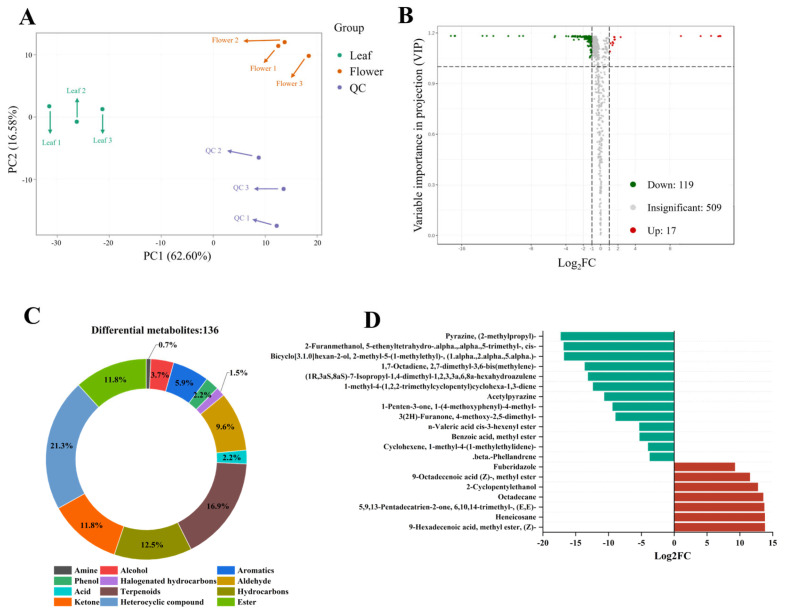
Screening of significantly differentially expressed volatile metabolites in yellowhorn leaves and flowers. (**A**) PCA scores plots; (**B**) volcano plot of volatile metabolites; (**C**) proportion of different classes of differentially expressed volatile metabolites; (**D**) differentially expressed volatile metabolites in the leaf and flower of yellowhorn.

**Figure 5 foods-13-01274-f005:**
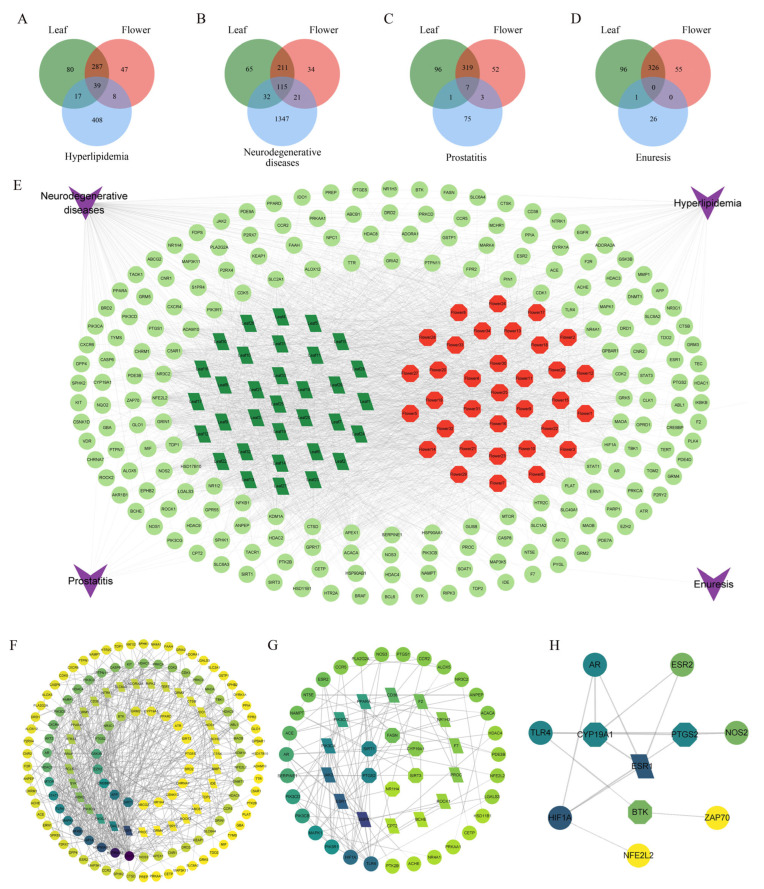
Network pharmacological analysis of non-volatile metabolites in yellowhorn leaves and flowers. (**A**–**D**) Venn diagrams of predicted targets of non-volatile metabolites and hyperlipidemia, neurodegenerative diseases, prostatitis, and enuresis targets; (**E**) “metabolite–target–disease” network; (**F**–**H**) PPI network diagram of potential targets for neurodegenerative diseases, hyperlipidemia, and prostatitis of non-volatile metabolites.

## Data Availability

The original contributions presented in the study are included in the article/Appendix A, further inquiries can be directed to the corresponding author.

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
