# Peer review of "Widely Targeted Metabolomics and Network Pharmacology Reveal the Nutritional Potential of Yellowhorn (Xanthoceras sorbifolium Bunge) Leaves and Flowers"

_foods, 2024, doi:10.3390/foods13081274_

Round 1

Reviewer 1 Report

Comments and Suggestions for Authors

Dear Authors,

I would like to congratulate you on the work you have prepared, which required a great deal of expertise and commitment on the part of the authors in preparing the experiments and compiling the results.

The paper is prepared carefully, the introductory part gives the reader sufficient information about the reasons for undertaking and carrying out the research described in the following section. The methodology of the research carried out is described in such detail that it can easily be reproduced by other researchers if necessary. Appropriate techniques for the identification of individual groups of compounds have also been used. The descriptions of the identified volatile and non-volatile components are very detailed within each group.

The figures enhance interpretation and are well prepared (legible), making the discussion of results and conclusions understandable to the reader. Literature items are properly cited and relevant to the studies discussed in this thesis.

My only comment relates to the supplementary section - table S2, where there is no information on what basis it was prepared and for what purpose, as I do not see a reference in the paper. However, this does not fundamentally affect my assessment.

Kind regards

Author Response

Response to Reviewer 1 Comments

I would like to congratulate you on the work you have prepared, which required a great deal of expertise and commitment on the part of the authors in preparing the experiments and compiling the results.

The paper is prepared carefully, the introductory part gives the reader sufficient information about the reasons for undertaking and carrying out the research described in the following section. The methodology of the research carried out is described in such detail that it can easily be reproduced by other researchers if necessary. Appropriate techniques for the identification of individual groups of compounds have also been used. The descriptions of the identified volatile and non-volatile components are very detailed within each group.

The figures enhance interpretation and are well prepared (legible), making the discussion of results and conclusions understandable to the reader. Literature items are properly cited and relevant to the studies discussed in this thesis.

My only comment relates to the supplementary section - table S2, where there is no information on what basis it was prepared and for what purpose, as I do not see a reference in the paper. However, this does not fundamentally affect my assessment.

Kind regards

Response: Thank you for your recognition of our work. The supplementary section Table S2 (now Table S3) shows the gene symbol of target proteins that interact with non-volatile differential metabolites in flowers and leaves. We have modified this table title to make it easier to understand. This analysis method is described in Section 2.6 (lines 133-142). Through the part (Section 3.6, lines 434-480), we hope to reveal the nutritional potential of the flowers and leaves of yellowhorn.

Reviewer 2 Report

Comments and Suggestions for Authors

Comments for authors

  1. The title lacks appeal due to excessive length and content.
  2. Further elaboration is needed on targeted metabolomics to provide clarity.
  3. The abstract should include a brief statement on the methodological techniques utilized in the study.
  4. Details regarding the detection of volatiles and non-volatiles should be incorporated by the authors.
  5. A thorough English editing of the entire manuscript is highly recommended.
  6. The rationale behind selecting compounds (3-7) in the terpenoids section, particularly for their predominance or biological activity, should be clarified. Additionally, it should be specified whether these compounds have been previously isolated from the species under study.
  7. Figure 2 should be removed upon completion of this section.
  8. PCA figures would be beneficial additions to the manuscript for better comprehension by readers.
  9. Figures need to be enhanced as the current resolution is insufficient, making text unclear.
  10. The authors should detail the methods used for the identification of both volatile and nonvolatile metabolites.
  11. Mass spectra should be provided by the authors.
  12. The tables presenting identified compounds lack crucial information such as molecular weight or fragmentation data for GCMS. The authors need to clarify how these compounds were identified.

Additionally, comprehensive tables containing detailed data on these metabolites, along with references for the identified compounds, should be included. Furthermore, the manuscript would benefit from the addition of more MS spectra. Addressing the methodology section and incorporating the suggested enhancements will greatly enhance the manuscript's suitability for publication.

Comments on the Quality of English Language
  1. A thorough English editing of the entire manuscript is highly recommended.

Author Response

Response to Reviewer 2 Comments

  1. The title lacks appeal due to excessive length and content.

Response: Thanks to your suggestion, we have revised the title (lines 1-4).

  1. Further elaboration is needed on targeted metabolomics to provide clarity.

Response: Detailed information of targeted metabolomics was added in line 11.

  1. The abstract should include a brief statement on the methodological techniques utilized in the study.

Response: We have added this brief introduction in the abstract (lines 11-12).

  1. Details regarding the detection of volatiles and non-volatiles should be incorporated by the authors.

Response: We thank you for raising the issue. We have revised the method according to your suggestions (lines 77-130).

  1. A thorough English editing of the entire manuscript is highly recommended.

Response: Thank you for your constructive comments on our manuscript. We revised the English editing of the manuscript.

  1. The rationale behind selecting compounds (3-7) in the terpenoids section, particularly for their predominance or biological activity, should be clarified. Additionally, it should be specified whether these compounds have been previously isolated from the species under study.

Response: Thank you for raising the issue. We have modified this part (lines 200-208).

In addition, we searched the relevant literature. There is no information to indicate that these components are found in the yellowhorn. Hence, these components have not been isolated from the yellowhorn. According to your comments, we have added this information (lines 206-208).

  1. Figure 2 should be removed upon completion of this section.

Response: Thank you for pointing this out. However, Figure 2 shows the relative content of non-volatile metabolites in yellowhorn leaves and flowers. This is important for understanding the composition and function of the flowers and leaves of yellowhorn. Therefore, we decided that Figure 2 should be retained.

  1. PCA figures would be beneficial additions to the manuscript for better comprehension by readers.

Response: The PCA figure has been added to Figure 1 (Line 159).

  1. Figures need to be enhanced as the current resolution is insufficient, making text unclear.

Response: We have checked and revised these Figures thoroughly according to your suggestions (Lines 159, 369, 408, and 481).

  1. The authors should detail the methods used for the identification of both volatile and nonvolatile metabolites.

Response: As the Reviewer suggested detailed methods for identifying volatile and non-volatile metabolites have been added to the manuscript (lines 122-128).

  1. Mass spectra should be provided by the authors.

Response: Considering the Reviewer’s suggestion, we have added the information to the supplementary material (Table S1, Figuer S1, S2, and S3).

  1. The tables presenting identified compounds lack crucial information such as molecular weight or fragmentation data for GCMS. The authors need to clarify how these compounds were identified.

Response: We have added this data to supplementary materials (Table S2), and more comprehensive UPLC-MS/MS and GC-MS information can be found in supplementary materials (Table S1).

Additionally, comprehensive tables containing detailed data on these metabolites, along with references for the identified compounds, should be included. Furthermore, the manuscript would benefit from the addition of more MS spectra. Addressing the methodology section and incorporating the suggested enhancements will greatly enhance the manuscript's suitability for publication.

Response: Thank you very much for your valuable comments and suggestions on our manuscript. Detailed metabolite data have been added to the supplementary materials (Table S1). And the mass spectra have been provided (Figure S1, S2, and S3)

Comments on the Quality of English Language

  1. A thorough English editing of the entire manuscript is highly recommended.

Response: Thank you very much for your suggestion, we have sought a professional agency to revise the English editing of the manuscript. Here we did not list the changes but marked in red in the revised paper.

Reviewer 3 Report

Comments and Suggestions for Authors

I read the work with great interest and learned a great deal about the metabolites produced by the leaves and flowers of yellowhorn (Xanthoceras sorbifolium Bunge) and their potential use for medicinal purposes and as food. However, I believe that the manuscript should be supplemented with the information indicated.

2.1. Materials: there is no information about the material studied. Is it known at least under what geographic and climatic conditions the yellowhorn trees from which the leaves and flowers used in the study were grown. This is important because the conditions in which the plant grows have a very strong influence on the metabolites it produces.

2.2.1. Sample preparation and extraction: The freeze-dried leaves and flowers were ground to powder - what were the average dimensions (or range of dimensions) of the powdered samples? Was a preliminary study done to determine the most favorable powdered sample/methanol solution ratio? Both the fineness of the sample and the sample/solvent ratio have a major impact on extraction efficiencies. The same applies to section 2.3.1. (lines 113-122).

Based on the information provided in sections 2.2.2. UPLC-ESI-MS/MS analysis and 2.3.2. GC-MS analysis – it is difficult to imagine how 948 and 976 non-volatile metabolites, as well as 638 and 636 volatile compounds, were identified, chemically structured and quantified. The authors should further explain to the reader how they obtained such detailed data, which they presented in Section 3. Results and Discussion (lines 162-175 and 378-392.

2.4. MS data processing and analysis: PCA and OPLS-DA were used to interpret the obtained results. However, there is no information on what specific figures were used to construct the matrix for PCA and OPLS-DA calculations. For this reason, I cannot understand why there are only 3 scores each in Figure 4A (PC1 and PC2) - my guess is that for leaves (L1, L2, L3) and flowers (F1, F2, F3). In the caption for Figure 4 and nowhere in the entire manuscript is there any information about what L1, L2, L3, F1, F2, and F3 mean.

Author Response

Response to Reviewer 3 Comments

I read the work with great interest and learned a great deal about the metabolites produced by the leaves and flowers of yellowhorn (Xanthoceras sorbifolium Bunge) and their potential use for medicinal purposes and as food. However, I believe that the manuscript should be supplemented with the information indicated.

Response: Thank you for your constructive comments on our manuscript. We revised the manuscript thoroughly according to your suggestions and marked in red.

2.1. Materials: there is no information about the material studied. Is it known at least under what geographic and climatic conditions the yellowhorn trees from which the leaves and flowers used in the study were grown. This is important because the conditions in which the plant grows have a very strong influence on the metabolites it produces.

Response: Thank you for the comment. In the study, yellowhorn is grown in Weifang (Longitude: 119.1, latitude: 36.62), China. We have added this information on lines 75-76.

2.2.1. Sample preparation and extraction: The freeze-dried leaves and flowers were ground to powder - what were the average dimensions (or range of dimensions) of the powdered samples? Was a preliminary study done to determine the most favorable powdered sample/methanol solution ratio? Both the fineness of the sample and the sample/solvent ratio have a major impact on extraction efficiencies. The same applies to section 2.3.1. (lines 113-122).

Response: In the study, the average dimensions of powdered samples were not measured. However, we found that the output particle size of the mixer mill (MM 400, Retsch) is about 5 μm. Therefore, we believe that the size of the sample powder is also in this range. We didn't do a preliminary study to determine the most favorable powdered sample/methanol solution ratio. The UPLC-MS sample pretreatment referred to the method of Chen et al [1]. The GC-MS sample pretreatment referred to the method of Wei et al [2].

References:

  1. Chen, W.; Gong, L.; Guo, Z.; Wang, W.; Zhang, H.; Liu, X.; Yu, S.; Xiong, L.; Luo, J. A Novel Integrated Method for Large-Scale Detection, Identification, and Quantification of Widely Targeted Metabolites: Application in the Study of Rice Metabolomics. Mol. Plant 2013, 6, 1769–1780, doi:10.1093/mp/sst080.
  2. Wei, G.; Tian, P.; Zhang, F.; Qin, H.; Miao, H.; Chen, Q.; Hu, Z.; Cao, L.; Wang, M.; Gu, X.; et al. Integrative Analyses of Nontargeted Volatile Profiling and Transcriptome Data Provide Molecular Insight into VOC Diversity in Cucumber Plants (Cucumis Sativus). Plant Physiol. 2016, 172, 603–618, doi:10.1104/pp.16.01051.

Based on the information provided in sections 2.2.2. UPLC-ESI-MS/MS analysis and 2.3.2. GC-MS analysis – it is difficult to imagine how 948 and 976 non-volatile metabolites, as well as 638 and 636 volatile compounds, were identified, chemically structured and quantified. The authors should further explain to the reader how they obtained such detailed data, which they presented in Section 3. Results and Discussion (lines 162-175 and 378-392).

Response: We have added the detailed identification method of non-volatile and volatile metabolites according to the Reviewer’s suggestion (lines 77-128). In addition, all metabolites of flowers and leaves of yellowhorn were presented in supplementary materials (Table S1).

2.4. MS data processing and analysis: PCA and OPLS-DA were used to interpret the obtained results. However, there is no information on what specific figures were used to construct the matrix for PCA and OPLS-DA calculations. For this reason, I cannot understand why there are only 3 scores each in Figure 4A (PC1 and PC2) - my guess is that for leaves (L1, L2, L3) and flowers (F1, F2, F3). In the caption for Figure 4 and nowhere in the entire manuscript is there any information about what L1, L2, L3, F1, F2, and F3 mean.

Response: Considering the Reviewer’s suggestion, we have revised the data processing and analysis (lines 122-128). The PCA and OPLS-DA were calculated using the data in Table S1. In addition, L1, L2, and L3 mean leaves, F1, F2, and F3 mean flowers. We have changed L1, L2, L3, F1, F2, and F3 to Leaf 1, Leaf 2, Leaf 3, Flower 1, Flower 2, and Flower 3 for better understanding (line 81).

Round 2

Reviewer 2 Report

Comments and Suggestions for Authors

 no other comments required.

Reviewer 3 Report

Comments and Suggestions for Authors

All my comments have been properly explained in the revised manuscript. The paper is suitable for publication in Foods.